# Fast Dark Signal Measurements of SVOM VT CCDs Using the Vertical Gradient of Dark Field Images

**Yue Pan [1,2], Xuewu Fan [1,*], Hui Zhao [1], Yulei Qiu [3], Wei Gao [1] and Jian Zhang [1]**

1   Xi'an Institute of Optics and Precision Mechanics, Chinese Academy of Sciences, Xi'an 710119, China; panyue@opt.cn (Y.P.); zhaohui@opt.ac.cn (H.Z.); gaowei@opt.ac.cn (W.G.); zhangjian@opt.ac.cn (J.Z.)
2   University of Chinese Academy of Sciences, Beijing 100049, China
3   National Astronomical Observatories, Chinese Academy of Sciences, Beijing 100012, China; qiuyl@nao.cas.cn
*   Correspondence: fanxuewu@opt.ac.cn

**Abstract:** This paper describes a fast technique for estimating the dark signals of the charge coupled devices (CCDs) of the visible telescope (VT) onboard the space multi-band variable object monitor (SVOM). It is based on the vertical gradient in the dark field images of the frame transfer CCDs. During the process of frame clear, exposure, frame transfer and readout, the characteristic of dark signal accumulation is analyzed firstly. Next, the linear fitting method is used to fit the signal level of the dark field image in the vertical direction, and the slope of the fitting line represents the dark signal factor. This technique only needs one dark field image and can be used for simple and efficient dark signal measurements of frame transfer CCDs. Besides, an experiment of detecting dark signals as a function of temperature based on the fast technique has been carried out. Making use of the Shockley-Hall-Read theory, two curve fitting formulas are adopted to the experimental results for VT Advanced Inverted Mode Operation (AIMO) CCD and VT Non-Inverted Mode Operation (NIMO) CCD respectively. The experimental results and the formulas are used to determine the optimal on-orbit cooling temperature of VT CCDs.

**Keywords:** charge coupled device (CCD); CCD characterization; dark signal; vertical gradient; SVOM VT

## 1. Introduction

Space multi-band Variable Object Monitor [1,2] (SVOM) is a proposed Chinese-French astronomical satellite, dedicated to the detection, localization and measurement of gamma-ray bursts (GRBs), while the Visible Telescope [3,4] (VT) is a visible and near-infrared instrument onboard the SVOM. There are two simultaneous channels in VT which is 400−650 nm blue channel and 650–1000 nm red channel [5,6]. The detector for the blue channel is an Advanced Inverted Mode Operation (AIMO), back-illuminated, basic processed E2V CCD42-80 device with a mid-band antireflection coating, while for the red channel is a Non-Inverted Mode Operation (NIMO), back-illuminated, basic processed E2V CCD42-80 device with an extended red coating manufactured on deep depleted silicon.

The dark signal [7] of a CCD is the signal output when no incident light is present, and it has an important influence on the detection signal-to-noise ratio and detection limit [8] of VT. During on-orbit observation of VT, the dark signal should be reduced as much as possible by cooling. Therefore, it is necessary to measure the dark signal temperature dependences of VT CCDs and determine the optimal cooling temperature.

The traditional method [9,10] of dark signal measurement needs to obtain at least two images, while one is a dark field image at long integration time, and the other one is a dark field image at short integration time or a bias image at zero integration time. The dark signal per unit time can be obtained by subtracting the two images, dividing the average signal of the result image by the difference of integration time, and multiplying by the system gain. For more accurate determination, multiple dark field images with

different integration time should be obtained, and the dark current is calculated by using the slope of the average signal of each image versus the integration time. However, when measuring the dark signal temperature dependence, the longer integration time and the longer waiting time for cooling temperature to stabilize will greatly affect the test efficiency. On the other hand, the dark signal is still accumulating in the readout process after frame transfer [11], because of the longer readout time of VT CCDs, about 20 s per frame at 100 kHz readout frequency. So the dark signals at different positions on the dark field image are different. Therefore, a fast and accurate dark signal measurement method based on the vertical gradient in the dark field image is proposed and applied.

In Section 2, based on the characteristics of frame clear, exposure, frame transfer and readout process of VT CCDs, a fast dark signal measurement technique based on the linear fit method is proposed. In Section 3, experiments of detecting dark signals as a function of temperature based on the fast technique have been carried out. Then two curve fitting formulas are adopted to the relationships between dark signals and temperatures for AIMO CCD and NIMO CCD respectively and meet a great agreement with the experimental data. Discussion and summary are provided in Sections 4 and 5.

## 2. Methods

The E2V CCD42-80 used in SVOM VT is shown in Figure 1, which is composed of two areas: sensitive area and storage area. The sensitive area is 2048 × 2048 pixels, and the storage area is 2048 × 2052 pixels. The extra four rows of pixels in the storage area are located at the junction of the storage area and the sensitive area, so as to prevent the light leakage. In addition, there are left and right simultaneous readout registers in the CCD, and each register has an extra 50 blank pixels.

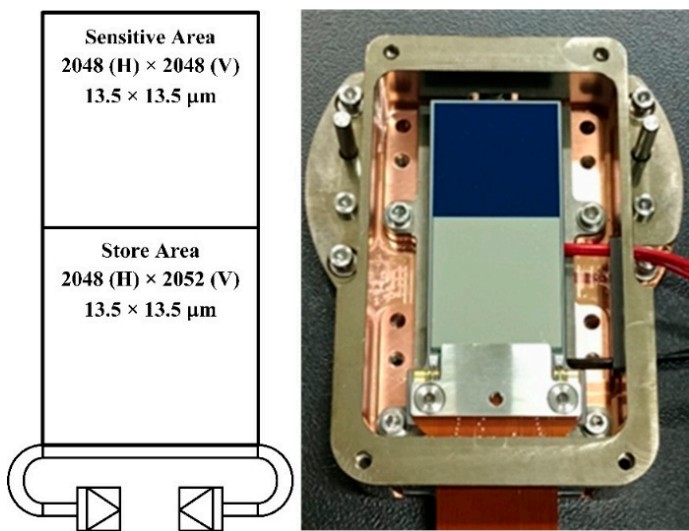

**Figure 1.** Schematic and physical drawings of E2V CCD42-80.

The driving timing mode of the CCD is called Clear Mode and contains four processes: frame clear, exposure, frame transfer and readout, as shown in Figure 2. In the frame clear process, the sensitive area transfers the useless charges to the storage area and clears itself, but the storage area will not be cleared. Then the sensitive area starts the exposure process. After that, in the frame transfer process, the image area transfers the effective charges to the storage area, and the storage area simultaneously drops the charges stored in the frame clear process. Finally, the readout process takes place. So, the dark signal should be calculated separately from the four processes.

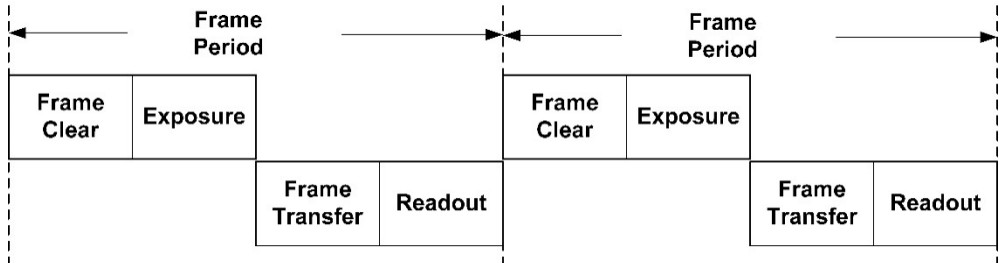

**Figure 2.** The Clear Mode driving timing of the CCD.

In the frame clear process, the sensitive area transfers the charge to the storage area at a rate of one row every $t_{row}$, where $t_{row} = 180$ μs/row for E2V CCD42-80, as shown in Figure 3. Therefore, the rows of the sensitive area near the storage area will collect more dark signal than that far away from the storage area, and the dark signal of each row grows in a linear trend, as:

$$S_{FC}(i_{row}) = (N_{FT} - i_{row} + 1) \cdot t_{row} \cdot S_{dark} \tag{1}$$

where $S_{FC}(i_{row})$ is the dark signal of *i-th* row collected during the frame clear process, $N_{FT}$ is the Number of transfers, $S_{dark}$ is the dark signal generated per second.

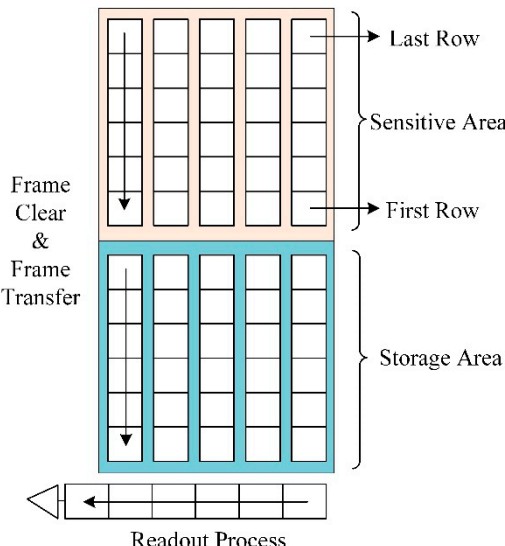

**Figure 3.** Frame clear, frame transfer and readout processes of the CCD.

In the exposure process, each of the rows collects the same dark signal, and the dark signal of *i-th* row collected during the exposure process $S_E(i_{row})$ at the integration time of $t_{int}$ can be expressed as:

$$S_E(i_{row}) = t_{int} \cdot S_{dark} \tag{2}$$

In the frame transfer process, the accumulated dark signal of each row is also the same, and the dark signal of *i-th* row collected during the exposure process $S_{FT}(i_{row})$ can be expressed as:

$$S_{FT}(i_{row}) = t_{row} \cdot S_{dark} \cdot N_{FT} \tag{3}$$

The final process is the readout. The data of the first row of pixels in the storage area are transferred to the horizontal readout register and read out from the left or right channel in turn at the rate of $v_{read} = 100{,}000$ pix/s@100 kHz. After the first row has been read out, the data of the second row will be transferred and the horizontal readout process will repeat. Because of the different readout time of each row of pixels, the dark signal

accumulated in each row is also different. The dark signal of *i-th* row collected during the readout process $S_R(i_{row})$ can be expressed as:

$$S_R(i_{row}) = (t_{row} + N_{pix}/v_{read}) \cdot S_{dark} \cdot i_{row} \tag{4}$$

where $N_{pix}$ is the number of columns readout from the left or right channel.

By joining Equations (1)–(4), the total dark signal of each row in the dark field image can be expressed as:

$$
\begin{aligned}
S_{dark,img}(i_{row}) &= S_{FC}(i_{row}) + S_E(i_{row}) + S_{FT}(i_{row}) + S_R(i_{row}) \\
&= (N_{FT}+1) \cdot t_{row} \cdot S_{dark} + t_{int} \cdot S_{dark} + t_{row} \cdot S_{dark} \cdot N_{FT} + N_{pix}/v_{read} \cdot S_{dark} \cdot i_{row}
\end{aligned}
\tag{5}
$$

where: $S_{dark,img}(i_{row})$ is the dark signal of the *i-th* row in the dark field image.

It can be seen from Equation (5) that the dark signal of each row increases linearly with the row number. Therefore, a signal value linear fitting method of dark field image is proposed to calculate dark signal. The signal value of the dark field image is averaged along the row direction to get 2048 average dark rows firstly. Then the diagram is drawn by the average signal value of each line and its line number, and the slope is $k_{dark}$. Thus, the dark signal generated per second can be expressed as:

$$S_{dark} = \frac{k_{dark}}{N_{pix}/v_{read}} \tag{6}$$

In the process of CCD dark signal measurements of SVOM VT, it is necessary to measure the dark signal temperature dependence. Based on the Shockley-Hall-Read theory, the relationship between the dark signal of CCD and temperature is given as [12]:

$$S_{dark} = C \cdot T^n \cdot e^{\frac{-E_g}{mK_BT}} \tag{7}$$

where $C$ is a constant, $T$ is the Kelvin temperature, $E_g \approx 1.12$ eV is the band gap energy of silicon material, $K_B$ is Boltzmann constant, $m$ and $n$ are coefficients, $n$ is generally between 1 and 3, $m$ is 1 for diffusion type dark signal and 2 for depletion type dark signal.

The dark signal generated in the CCD can be divided into two parts: the dark signal generated on the surface of the detector silicon material $S_{dark,surface}$ and the dark signal generated inside the detector silicon material $S_{dark,bulk}$.

For red channel NIMO CCD, the surface dark signal is two orders of magnitude higher than the depletion and diffusion [13,14], so the surface dark signal plays a leading role. Since the surface dark signal is mainly a depletion type, $m = 2$ and $n = 3$ are adopted in Equation (7). A constant term $C_2$ is introduced to represent the limiting dark signal caused by trap charge or system light leakage, then:

$$S_{dark,NIMO} = C_1 \cdot T^3 \cdot e^{\frac{-6400}{T}} + C_2 \tag{8}$$

For the blue channel AIMO CCD, the device is operated in an Inverted Mode (also known as Multi-Phased Pinned, MPP). In this mode, the clock voltages are driven more negative, which makes the surface interface full of holes, and inhibits the generation of electron-hole pairs [15]. So that the surface dark signal is suppressed, and only the internal dark signal exists, including both diffusion type and depletion type. So $m = 1~2$ and $n = 1~3$. A constant term $C_2$ is also introduced to represent the limiting dark signal caused by trap charge or system light leakage, then:

$$S_{dark,AIMO} = C_1 \cdot T^n \cdot e^{\frac{-12800}{mT}} + C_2 \tag{9}$$

Based on the dark signal temperature dependence expressed as Equations (8) and (9), the dark signals at multiple temperatures have been measured and the undetermined parameters can be fitted by the least square method.

## 3. Experiments and Results

Traditional methods of measuring the dark signal temperature dependence need to obtain both bias images and dark field images at each temperature. So, temperature control and image acquisition must be carried out at each temperature. Thus, the test workload is large and the efficiency is low. According to the vertical signal value linear fitting method of dark field image proposed above, only one dark field image is needed to calculate the dark signal. Therefore, a fast method for measuring dark signals as a function of temperature is proposed.

The dark signal measurements of the CCD detectors have been carried out on the SVOM VT CCD test bench [6] in the laboratory shown in Figure 4. The main characteristics of the CCDs, including bias field images, dark field images, flat field images, system gain, readout noise, quantum efficiency, linearity, and so on, have been tested on the test bench to provide an objective assessment on the performance of the CCD detectors. The main equipment used for dark signal detection contains the vacuum refrigeration tank and the electronic system. The special vacuum refrigeration tank can provide a temperature range of $-170\sim+100$ °C and vacuum below $1 \times 10^{-3}$ Pa, which fully meets the working requirements of the CCD detectors. The specific measurement procedure is shown as following steps:

(1) The vacuum refrigeration tank starts the refrigeration process and sets the cooling rate of 1~2 °C/min, as well as the computer automatically records the temperature versus time data simultaneously;

(2) During the refrigeration process, the dark field image of 1 s integral time is acquired continuously, and the computer automatically records the dark field image versus time data simultaneously;

(3) Dark signals as a function of time are calculated by the vertical signal value linear fitting method;

(4) The dark signals as a function of temperature data are obtained by combining the data from step (1) and step (3).

(5) The measured data of dark signals as a function of temperature is brought into Equations (5) and (6), and the undetermined coefficient is fitted by the least square method.

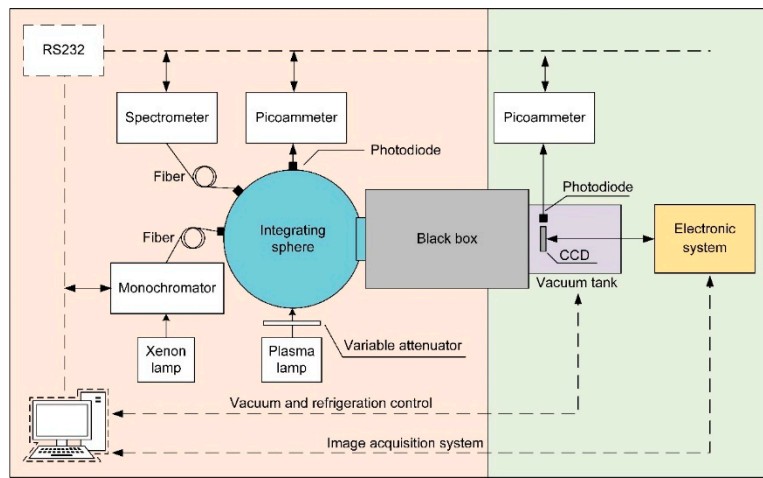

**Figure 4.** Structure diagram of the SVOM VT CCD test bench.

An image of a typical dark signal test result is shown in Figure 5, while the temperature is $-30$ °C, and the CCD is the red channel NIMO CCD42-80. There is an obvious gradient in the dark field image as shown in Figure 5, which meet a great agreement with the simulation Equation (5). The dark signal can be calculated by Equation (6), which is $S_{dark} = 10.93DN/\text{pix}/\text{s}$. In addition, the dark signal in electrons can be obtained by multiplying $S_{dark}$ by the system gain of the CCD.

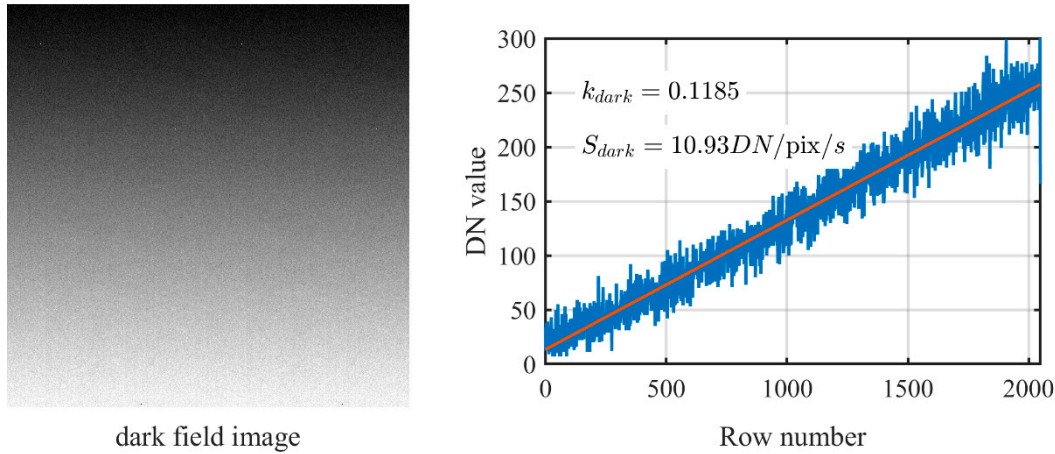

dark field image

Row number

**Figure 5.** Dark signal measurement results of red band NIMO CCD at −30 °C.

The whole test and fitting results of the dark signals as a function of temperature for red channel NIMO CCD are shown in Figure 6. The results show that the measured data and fitting equation can match very well with the determination coefficient R-squared = 0.9997, and the fitting equation in the unit of electrons converted by the gain of 1.65e/DN is shown in the Equation (10).

$$S_{dark,NIMO} = 3.28 \times 10^5 \cdot T^3 \cdot e^{\frac{-6400}{T}} + 0.03643 \tag{10}$$

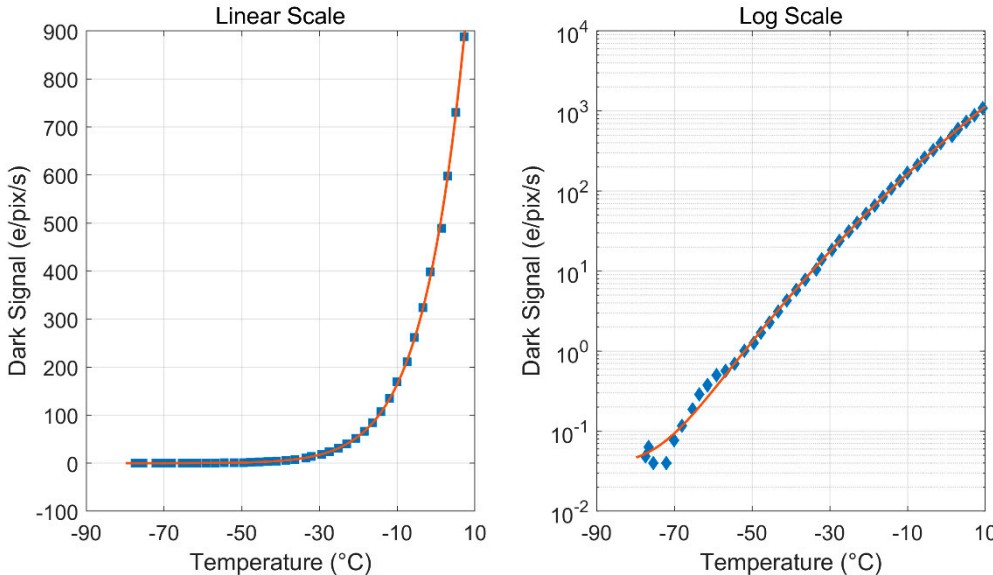

**Figure 6.** Dark signals as a function of temperature of red channel NIMO CCD.

The whole test and fitting results of dark signals as a function of temperature for blue channel AIMO CCD are shown in Figure 7. The results show that the measured data and fitting equation can match well with the determination coefficient R-squared = 0.9963, and the fitting equation in the unit of electrons converted by the gain of 1.45e/DN is shown in the Equation (11).

$$S_{dark,AIMO} = 1.412 \times 10^4 \cdot T^3 \cdot e^{\frac{-6937}{T}} + 0.0022 \tag{11}$$

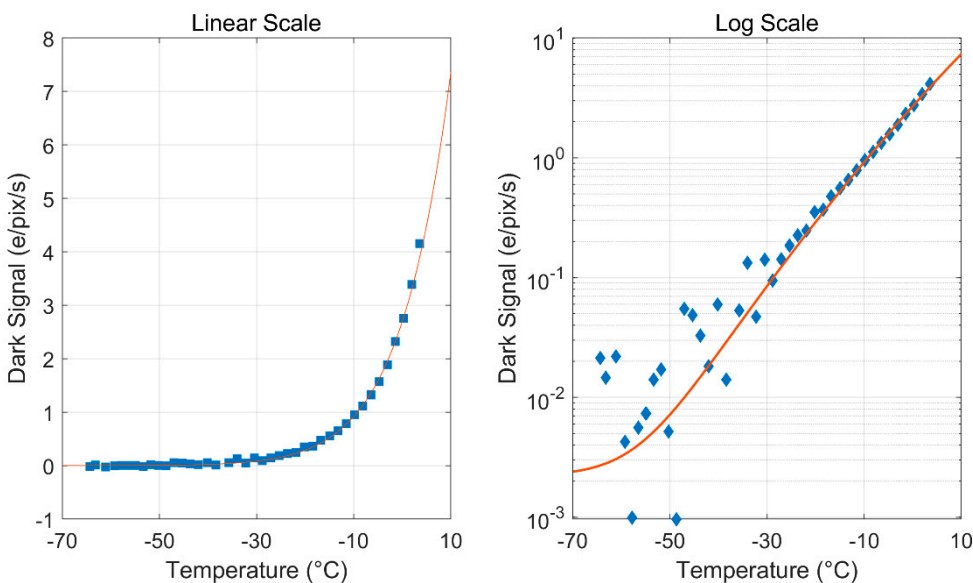

**Figure 7.** Dark signals as a function of temperature of blue channel AIMO CCD.

### 4. Discussion

From the fitting diagram in the logarithmic domain in Figure 6 of NIMO CCD, it can be seen that the test results and fitting results almost match perfectly in the range of −55~10 °C, while there is a little deviation in the range of −80~−55 °C. The main reason is that the dark signals at those temperatures are too low, which are lower than 1e/pix/s. When the temperature decreases to −70 °C, the dark signal is less than 0.1e/pix/s, and the dark signal changes little with the decrease in temperature below −70 °C. Therefore, the final on-orbit cooling temperature of the NIMO CCD is set to −75 °C.

On the other hand, the dark signal of AIMO CCD is much lower than that of NIMO CCD, with a difference of about two orders of magnitude. The main reason is that AIMO CCD suppresses the surface dark signal. From the fitting diagram in the logarithmic domain Figure 7 of AIMO CCD, it can be seen that the test results and fitting results match well in the range of −30~10 °C, while there is a certain deviation in the range of −80~−30 °C. The main reason is that the dark signals at these temperatures is extremely low, ranging from 0.001 to 0.1e/pix/s, so the dark signal collected in the last row of the image is less than one electron more than that in the first row, which is difficult to measure accurately. Fortunately, the linear part of the data is enough to fit the formula. When the temperature decreases to −40 °C, the dark signal is less than 0.01e/pix/s, and the dark signal changes little with the decrease in temperature below −40 °C. Therefore, considering the on-orbit operation and radiation damage [16], the final cooling temperature of the AIMO CCD is set to −65 °C.

In addition, we compared the fitting equations and test methods with those in the E2V datasheet [14]. For the red channel NIMO CCD, the fitting equations are exactly the same. We can simply measure the dark current at −25 °C, and then use the equation provided by E2V to calculate the dark current at lower temperature. However, for the blue channel AIMO CCD, there are some differences, where the fitting equation is $S_{dark,AIMO} = CT^3 e^{-\frac{9800}{T}}$ in the E2V datasheet [14]. For AIMO CCD, different device types will lead to the change of dark signal equation. Therefore, it is necessary to obtain dark signal temperature dependence by the actual measurement of the dark signal at different temperatures. In the references [15], the author measured the dark current at more than10 different temperatures using the traditional method, which was a large workload. Especially in low temperature, it needed to obtain dark field images at long integration time, and it needed to ensure the stability of the temperature controller during the integration period. With our method instead, measurements can be very fast. In the temperature range of −70 °C to 0 °C, we

can collect more than 80 data points within 40 min, which improves the fitting accuracy. In addition, the acquisition of each dark field image does not require the temperature stability. When the temperature is changed by 2 °C per minute, there will be a change of about 0.7 °C during each acquisition, which will bring about a systematic error of about 1%. The error is acceptable, and can be further reduced by setting the temperature ramping of the vacuum cooling system to 1 °C or lower per minute.

## 5. Summary

This paper describes the characteristics of the readout process of VT CCDs and proposed a fast dark signal measurement technique based on the linear fitting method. Then an experiment of detecting dark signals as a function of temperature based on the fast technique has been carried out. Two curve fitting formulas obtained by the analysis of dark signal sources are adopted to the dark signal temperature dependence for AIMO CCD and NIMO CCD respectively and meet a great agreement with the experimental data. According to the test and fitting results, the on-orbit cooling temperatures of VT CCDs are determined as AIMO −65 °C and NIMO −75 °C respectively. The technique proposed in this paper can be used for simple and efficient dark signal measurements of CCDs.

**Author Contributions:** Conceptualization, Y.P.; methodology, Y.P.; software, H.Z.; validation, X.F.; investigation, Y.Q.; resources, J.Z.; data curation, W.G.; writing—original draft preparation, Y.P.; writing—review and editing, Y.P. All authors have read and agreed to the published version of the manuscript.

**Funding:** This research was funded by the National Natural Science Foundation of China Grant No.61107008 and Grant No.61105017.

**Institutional Review Board Statement:** Not applicable.

**Informed Consent Statement:** Not applicable.

**Data Availability Statement:** Not applicable.

**Acknowledgments:** The authors thank the outstanding work of the SVOM VT team from Xi'an Institute of Optics and Precision Mechanics of CAS, National Astronomical Observatories of CAS, and Innovation academy for microsatellites of CAS.

**Conflicts of Interest:** The authors declare no conflict of interest.

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
