# Peer review of "Fast Dark Signal Measurements of SVOM VT CCDs Using the Vertical Gradient of Dark Field Images"

_photonics, doi:10.3390/photonics8040132_

Round 1

Reviewer 1 Report

The paper describes a method of measuring the dark current in CCDs using a single readout, using the dark signal integration in the store section of the CCD. After the store section has been fast-cleared of charge, subsequent normal (slow) readout results in a dark signal ramp in the parallel direction. This is well known and can be used in virtually any CCD, with or without a store section. 

I see several major issues which need addressing. 

MA1: p.1 line 40: the method described here is not what is normally used, because a bias image (without dark current) is generally not available. Instead, the usual method is to take several  (two or more) images at different integration times and to plot the signal, averaged over many pixels, vs the integration time. The slope of the data is the dark current. Only a small part of the CCD can be read out and the rest fast-dumped, so that to speed up the process. 

MA2: p. 1 line 44: The authors say that 1000 seconds may not be long enough to measure the dark current and yet their measurement is taking about 21 seconds. Can you clarify this?

MA3: p. 2 line 68: The process described here assumes that the CCD has initially zero charge in the image area, which is not true. Dark signal from the previous readout or frame transfer will also be present in the image section. As the signal is read out from the store area, dark current will integrate under the image area. What do you do with that charge? 

The way I think the method should work is to first do a fast clear of both image and store areas by clocking Nrow = 4100 rows without reading them. The dark signal accumulated in the CCD will be a ramp with the maximum signal in the first row near the readout register. The stored signal is described as Si = (Nrow - i +1) trow Idc  where i is the row number. Subsequent slow readout adds dark current integrated during the serial clocking and the output signal becomes

Si = (Nrow - i +1) trow Idc + i tread Idc 

Therefore, the slope dSi/di = (tread - trow)Idc = (Nser/fser)Idc   

This is slightly higher than (3) due to the subtraction of trow from tread

MA4: The method measures the dark current generated in the store area, which may be different from the dark current in the image area. Can the authors measure both, e.g. by reading the whole device, to confirm if this is the case? 

MA5: page 5: With the temperature ramping at 2 °C per minute the temperature will change by 0.73 °C during the measurement. Is that taken into account?

MA6: Can you explain Figure 4(a)? There are two ramps and two sections (the two halves of the serial register?). Please add scales and labels. 

MI1: Please explain "basic processed" with regard to CCD42-80

MI2: it is better to use "signal value" instead of "gray value"

MI3: p.1 line 38 and elsewhere: please use "operating temperature" or "cooling temperature" instead of "refrigeration temperature". 

Reviewer 2 Report

The paper proposes a method to measure the dark signal of a CCD in a more efficient way by using the vertical gradient linear fitting method. The proposed method efficiently measures the dark signal across temperature. I have several comments as follow:

1) In line 107, the parameter N is not found in equation 4. 

2) In line 119, the authors mentioned about the AIMO surface dark signal is suppressed, but the suppression mechanism is not discussed in the paper. 

3) From the results in Figures 5 and 6, the nonlinearity occurs when the dark signal rate reaches a certain level. I think more discussion is necessary to explain the reason for the nonlinearity. What is the conversion gain? How much is 1e- in terms of DN? What is the ADC resolution used? I suggest the authors include these points in the discussion. 

4) I would suggest the authors be consistent in the unit used, for example, Figures 5 and 6 use Kelvin for temperature, but the texts that reference the figures use degree Celcius. 

Round 2

Reviewer 1 Report

I would like to thank the authors for revising the paper and for clarifying the experimental method. In particular adding Figure 2 has helped greatly. 

There are still a few minor issues that I feel need to be addressed. 

  1. The authors talk about instabilities in the readout as one of the reasons for using a single image to derive the dark current. Such instabilities occur between images, but could occur during taking of an image. Have those been observed? 
  2.   page 2 line 46: the conversion gain -> by the conversion gain; occasions -> determination
  3. I would say that getting a negative pixel signal is not an issue because we are only interested in the slope, i.e. signal versus time. 
  4. page 2 line 74: CCD42-80 has 50 pre-scan (blank) elements according to the datasheet. The overscans are not physical pixels. 
  5. page 2 line 80: clear -> clears
  6.  page 3 line 88: "the sensitive area transfers the charge to the storage area at a rate of one row every trow, where ..."
  7. page 4 line 130: remove "undetermined"
  8. page 4 line 137 and 146: internal dark signal -> depletion and diffusion dark signal
  9. page 4 line 136 and 142: NIMO and AIMO have been defined on page 1, remove the re-definition. 
  10. page 5: the authors should mention the analysis on the temperature drift during CCD readout and the error it causes, as outlined in the response to the comments.
  11. page 5 line 176: Is the 1 second integration time needed, and can it be zero? 
  12. page 5 line 186: Figure 4 -> Figure 5. 
  13. page 5 line 191: you mean the system gain (e-/DN), not the conversion gain of the CCD (measured in µV/e-)
  14. References 14 and 15 seem incomplete - the name of the source is missing. 
